# Effect of Osimertinib on CTCs and ctDNA in EGFR Mutant Non-Small Cell Lung Cancer Patients: The Prognostic Relevance of Liquid Biopsy

**DOI:** 10.3390/cancers14061574

**Published:** 2022-03-19

**Authors:** Galatea Kallergi, Emmanouil Kontopodis, Aliki Ntzifa, Núria Jordana-Ariza, Niki Karachaliou, Evangelia Pantazaka, Haris A. Charalambous, Amanda Psyrri, Emily Tsaroucha, Ioannis Boukovinas, Anna Koumarianou, Dora Hatzidaki, Evi Lianidou, Vassilis Georgoulias, Rafael Rosell, Athanasios Kotsakis

**Affiliations:** 1Department of Biology, Division of Genetics, Cell and Developmental Biology, University of Patras, 26500 Patras, Greece; galkallergi@gmail.com (G.K.); evapantazaka@upatras.gr (E.P.); 2Department of Medical Oncology, “Venizelio-Pananio” General Hospital of Heraklion, 71500 Heraklion, Greece; kontopodise@gmail.com; 3Analysis of Circulating Tumor Cells Lab, Department of Chemistry, University of Athens, 15772 Athens, Greece; alntzi@chem.uoa.gr (A.N.); lianidou@chem.uoa.gr (E.L.); 4Lab of Analytical Chemistry, Department of Chemistry, University of Athens, 15772 Athens, Greece; 5Laboratory of Oncology, Pangaea Oncology, Quiron Dexeus University Hospital, 08028 Barcelona, Spain; njordana@panoncology.com (N.J.-A.); niki.karachaliou@merckgroup.com (N.K.); rrosell@iconcologia.net (R.R.); 6Oncology Center of Bank of Cyprus, Department of Medical Oncology, Nicosia 2012, Cyprus; haris.charalambous@bococ.org.cy; 7Medical Oncology Unit, 2nd Department of Internal Medicine, “ATTIKON” General Hospital of Athens, 12462 Athens, Greece; psyrri237@yahoo.com; 87th Department of Pulmonary Diseases, “SOTIRIA” General Hospital of Athens, 11527 Athens, Greece; emilygeola@yahoo.gr; 9Department of Medical Oncology, BIOCLINIKI Hospital, 11524 Thessaloniki, Greece; ibouk@otenet.gr; 10Hematology-Oncology Unit, Fourth Department of Internal Medicine, Attikon Hospital, Medical School, National and Kapodistrian University of Athens, 12462 Athens, Greece; akoumari@yahoo.com; 11Hellenic Oncology Research Group (HORG), 1st Department of Medical Oncology, Metropolitan General Hospital, 15562 Athens, Greece; dorachat@med.uoc.gr (D.H.); georgulv@otenet.gr (V.G.); 12Department of Medical Oncology, University General Hospital of Larissa, Mezourlo, 41500 Larissa, Greece

**Keywords:** osimertinib, CTCs, ctDNA, EGFR mutant, NSCLC

## Abstract

**Simple Summary:**

Osimertinib has become the standard of care for the first-line treatment of *EGFR*-mutant NSCLC patients. The aim of this current translational research study was to assess the clinical relevance of liquid biopsy in 47 patients receiving osimertinib. Effects on circulating tumor cells (CTCs) and plasma-DNA (ctDNA) were investigated before, after one treatment cycle, and at the end of treatment. ctDNA and CTCs decreased after one treatment cycle, but increased at the end of treatment. The detection of ctDNA before and after one treatment cycle was associated with shorter progression-free and overall survivals (PFS and OS), whereas ctDNA clearance after one treatment cycle resulted in a significantly longer PFS and OS. ctDNA at baseline emerged as an independent predictor of shorter PFS. Thus, changes in liquid biopsy status (CTCs, ctDNA) during osimertinib treatment can be used as a tool for treatment efficacy.

**Abstract:**

Introduction: Liquid biopsy is a useful tool for monitoring treatment outcome in solid tumors, including lung cancer. The relevance of monitoring CTCs and plasma ctDNA as predictors of clinical outcome was assessed in EGFR-mutant NSCLC patients treated with osimertinib. Methods: Forty-seven EGFR-mutant NSCLC patients who had progressed on prior first- or second-generation EGFR inhibitors were enrolled in the study and treated with osimertinib, irrespective of the presence of the T790M mutation in the primary tumor or the plasma. Peripheral blood was collected at baseline (*n* = 47), post-Cycle 1 (*n* = 47), and at the end of treatment (EOT; *n* = 39). CTCs were evaluated in 32 patients at the same time points (*n* = 32, *n* = 27, and *n* = 21, respectively) and phenotypic characterization was performed using triple immunofluorescence staining (CK/VIM/CD45). Results: Osimertinib resulted in an ORR of 34% (2 CR) and a DCR of 76.6%. The median PFS and OS values were 7.5 (range, 0.8–52.8) and 15.1 (range, 2.1–52.8) months, respectively. ctDNA was detected in 61.7%, 27.7%, and 61.5% of patients at baseline, post-Cycle 1, and EOT, respectively. CTCs (CK+/CD45-) were detected in 68.8%, 48.1%, and 61.9% of patients at the three time points, respectively. CTCs expressing both epithelial and mesenchymal markers (CK+/VIM+/CD45-) were detected in 56.3% and 29.6% of patients at baseline and post-Cycle 1, respectively. The detection of ctDNA at baseline and post-Cycle 1 was associated with shorter PFS and OS, whereas the ctDNA clearance post-Cycle 1 resulted in a significantly longer PFS and OS. Multivariate analysis revealed that male sex and the detection of ctDNA at baseline were independent predictors of shorter PFS (HR: 2.6, 95% C.I.: 1.2–5.5, *p* = 0.015 and HR: 3.0, 95% C.I.: 1.3–6.9; *p* = 0.009, respectively). Conclusions: The decrease in both CTCs and ctDNA occurring early during osimertinib treatment is predictive of better outcome, implying that liquid biopsy monitoring may be a valuable tool for the assessment of treatment efficacy.

## 1. Introduction

Non-small cell lung cancer (NSCLC) harboring somatic mutations in the tyrosine kinase domain of epidermal growth factor receptor (EGFR) represents a molecular subgroup of NSCLC requiring treatment with EGFR tyrosine kinase inhibitors (TKIs) [1,2,3,4,5,6,7,8]. In almost 60% of *EGFR*-mutant NSCLC patients, acquired resistance to first- and second-generation EGFR TKIs has been attributed to the emergence of the exon 20 *EGFR* T790M mutation [9,10,11]. The third-generation EGFR TKI osimertinib was initially approved for the treatment of patients with *EGFR* T790M-positive NSCLC who had progressed on prior EGFR TKIs [12,13]. However, osimertinib has now become the standard of care for the first-line treatment of *EGFR*-mutant NSCLC [8,14] and it was recently approved for use in the adjuvant setting following tumor resection in patients with *EGFR*-mutant NSCLC [15].

*EGFR*-mutant NSCLC is characterized by tumor heterogeneity that, under treatment pressure, may lead to the emergence of tumor clones with additional genetic alterations including *MET* or *HER2* amplification, *PIK3CA* mutations, or with histological transformation to small cell lung cancer (SCLC) and the epithelial-to-mesenchymal transition (EMT). All these genetic changes have been associated with resistance to EGFR TKIs [15,16,17,18]. Considering that tumor tissue is not always available for the detection of resistance mechanisms, liquid biopsy (LB) including circulating tumor cells (CTCs) and circulating tumor DNA (ctDNA), is an alternative approach [19,20,21]. CTCs represent a heterogeneous cell population [19]. CTCs often undergo EMT during their migration and harbor a mesenchymal phenotype [18,19,20,21] that cannot be recognized by epithelial marker-based detection assays such as the CellSearch platform [21,22,23,24,25,26,27,28]. Our group has previously reported that a substantial proportion of CTCs in cancer patients may express stem cell and EMT markers, such as aldehyde dehydrogenase 1 (ALDH1), twist, and vimentin (VIM) [29,30]. Therefore, the high prevalence of CTCs with a mesenchymal phenotype must be taken into consideration for the detection of CTCs in NSCLC patients. 

In NSCLC patients, the detection of CTC before treatment is associated with shorter progression-free survival (PFS) and overall survival (OS) [31,32]. In *EGFR*-mutant NSCLC patients, the genotyping of CTCs and tumor biopsies gave comparable results [31]. The detection of *EGFR* mutations in ctDNA from NSCLC patients has proved to be complementary to tissue biopsy in clinical practice and is correlated with both the tumor baseline lesion size and response to EGFR TKIs [33,34]. 

The aim of this translational research study was to investigate the effect of osimertinib on the changes in LB (both CTCs and ctDNA) status in patients with *EGFR*-mutant NSCLC and to assess the clinical relevance of those changes.

## 2. Materials and Methods

### 2.1. Study Design

Patients with histologically documented *EGFR*-mutant NSCLC and disease progression on first- or second-generation EGFR TKIs were treated with osimertinib and the changes in CTCs and ctDNA (LB status) were assessed during treatment. Patients had to fulfill the standard clinical study inclusion and exclusion eligibility criteria (Materials and Methods). Patients were treated in the Hellenic Oncology Research Group’s (HORG) collaborative centers and the study was approved by the Ethics Committees and the Institutional Review Boards of the participating hospitals, the National Ethic Committee (no: 35/00-03/16), the National Drug Organization (no: IS 28/16), and registered in the clinicaltrials.gov platform (number: NCT02771314, registration date: 13 May 2016) and EudraCT (number: 2016-001335-12, registration date: 13 April 2016). All patients provided written informed consent for participation in the study. 

### 2.2. Blood Samples

Peripheral blood (20 mL in EDTA) was obtained before the administration of osimertinib (baseline sample; Pre), after one month of treatment (Post-1 sample), and at the end of treatment (EOT sample), which corresponded to the time of disease progression. All blood samples were obtained from the middle of vein puncture, after the first 5 mL of blood was discarded to avoid contamination with epithelial cells from the skin. A 10 mL sample of blood was used for the capture of CTCs using the ISET-based filtration system; the sample was centrifuged at 2500 rpm for 10 min, the plasma was removed, and aliquots of 2 mL were stored at −80 °C until use. 

### 2.3. ISET Isolation Platform and Immunofluorescence Staining 

CTCs were isolated using the ISET system according to the manufacturer’s instructions. Briefly, 10 mL of peripheral blood were diluted 1:10 in ISET buffer (Rarecells, Paris, France) for 10 min at RT. Diluted samples were filtered using depression tab adjusted at 10 KPa. Membranes were dried for 2 h at RT and stored at −20 °C. Each spot on the membrane was used for identification of CTCs after IF staining. 

CTCs from patients and cells from lung cancer cell lines, used as controls, were evaluated for cytokeratin (CK)/CD45 expression and for CK/Vimentin (VIM)/CD45 expression by double and triple IF staining, respectively, using the appropriate antibodies as previously described [31]; samples were evaluated using confocal laser scanning microscopy (details of the antibodies used and the immunofluorescence (IF) staining are presented in Materials and Methods. The cytomorphological criteria described by Meng et al. [35] were used for the characterization of a cell as a CTC.

### 2.4. ctDNA Isolation and PNA-Q-PCR Assay for Mutation Testing

ctDNA was extracted from two aliquots of 2 mL of plasma using the QIAsymphony^®^ DSP Virus/Pathogen Midi Kit and a QIAsymphony robot (Qiagen, Amtsgericht Düsseldorf, HRB 45822, Hilden, Germany), following the manufacturer’s instructions. The final elution volume was 30 µL per 2 mL aliquot. *EGFR* sensitizing (exons 19, 21) and resistance mutations (T790M and C797S) were detected using a quantitative real-time PCR (Taqman^®^, Rue Bois Saint-Jean 5, Rue du Bois Saint-Jean 14, 4102, Seraing, Belgium) assay in the presence of a PNA clamp (Eurogentec, Rue Bois Saint-Jean 5, Rue du Bois Saint-Jean 14, 4102, Seraing, Belgium), which inhibits the amplification of the wild-type alleles, as previously described [36]; the details for the assay are presented in Materials and Methods. 

### 2.5. Statistical Analysis

This was a prospective, non-randomized, multi-center, translational research study. No formal statistical sample size calculation was performed. Progression-free survival (PFS) was defined as the time elapsed between the start date of treatment and the date of clinical or radiological progression or death from any reason. Overall survival (OS) was defined as the time elapsed between the start date of treatment until the date of death from any reason or the date of last follow-up. Qualitative factors were compared by Pearson’s Chi-square test or Fisher’s exact test, whenever appropriate. Differences in positivity rates were assessed using the McNemar test and differences in terms of continuous variables were assessed by the non-parametric Wilcoxon test. PFS and OS for all patients were estimated using the Kaplan–Meier analysis and the comparisons were computed with the log-rank test. Associations between prognostic factors and PFS or OS were examined using Cox proportional hazards regression models. All statistical tests were two-sided and *p*-values < 0.05 were considered statistically significant. Data were analyzed using the SPSS statistical software, version 22.0 (SPSS Inc., Chicago, IL, USA). 

## 3. Results

### 3.1. Study Population

In total, 50 *EGFR*-mutant NSCLC patients who had progressed on prior first- or second-generation EGFR TKIs, were registered and 48 of them were treated since two of them withdrew their consent. LB status was assessed in 47 patients since one patient had no blood sampling at baseline for technical reasons (CONSORT Diagram; Appendix A). The baseline characteristics of the patients are summarized in Table 1. The median age was 66 years (range: 43–87), 34 (72.3%) were women, all had adenocarcinoma, and 57.4% had a PS (ECOG) of 0. Molecular analysis of the primary tumor, which was performed in the initial tissue sample used for diagnosis, is presented in Appendix A. The tumoral detection of the *EGFR* exon 20 T790M mutation was detected in 14 (29.8%) patients; moreover, three additional patients harbored the T790M mutation in the baseline blood sample but not in the primary tumor. Twenty-three (48.9%) patients had received more than two prior lines of treatment with either first- or second-generation EGFR TKIs and/or chemotherapy. Thirteen (27.7%) and twelve (25.5%) patients had liver and CNS metastases, respectively, and the median number of involved sites was three (range, 1–5). 

### 3.2. Efficacy 

A summary of the efficacy results is presented in Table 1. Briefly, there were 2 patients with complete response (CR) and 14 patients with partial response (PR) for an overall objective response rate (ORR) of 34.0% (95% C.I., 20.5–47.6%) and a disease control rate (DCR) of 76.6% (95% C.I., 64.5–88.7%). For a median follow-up period of 41.9 months, the median PFS was 7.5 months (95% C.I., 6.0–9.0; range, 0.8–52.8), the median OS was 15.1 months (95% C.I., 10.8–19.4; range, 2.1–52.8 months), and the 1-year survival rate was 69.8%. Four patients were alive and without disease progression for 53, 45, 42, and 36 months, whereas 35 (74.5%) had died by the time of analysis. Adverse events were as expected and are presented in Appendix A.

### 3.3. CTC and ctDNA Status before Study Treatment (Pre Sample) 

Adequate biological material for the assessment of CTC status at baseline was available in 32 (68.1%) patients; the assessment of 15 patients failed for technical reasons. CTC could be detected in 22 (68.8%) patients by double IF staining [total number of CTCs = 56 with a median of 2.5/mL of blood (range, 1–5/mL)]. Triple IF staining in these patients revealed a heterogeneous population of CTCs based on the expression of CK (CK^+^) and VIM (VIM^+^). In 12 (37.5%) patients, CTCs had exclusively an EMT (CK^+^/VIM^+^/CD45^-^; *n* = 8) or epithelial (CK^+^/VIM^-^/CD45^-^; *n* = 4) phenotype, whereas in 10 (31.2%) patients, both subpopulations could be detected (Table 2). Representative images of triple IF staining are presented in Appendix A. 

ctDNA molecular profiling of baseline samples was available for all patients and *EGFR* mutations were detected in 29 (61.7%) patients (Table 3). There was no correlation between the detection of CTCs and ctDNA in the baseline samples, since seven (21.9%) patients had detectable ctDNA in their plasma, but not CTCs in the same blood drawn (*p* = 0.703). 

### 3.4. CTC and ctDNA Status after One Month of Study Treatment (Post-1 Sample)

CTCs were also assessed by double IF staining after the first cycle in 39 (83%) patients, and 20 (51.3%) patients had detectable CTCs [total number of CTCs = 35 (median number: 1/mL of blood; range, 1–13)]. There was a statistically significant decrease in the total number of CTCs at the Post-1 time point compared to the baseline sample (*p* = 0.037; Wilcoxon test). ISET filters from 27 (69.2%) patients were also available for triple IF staining and in 23 (85.2%) patients, CTCs detection was available in both time points (Pre and Post-1). Eight (34.8%) patients with CTCs detectable in the baseline sample had CTCs non-detectable in the Post-1 sample, whereas 3 (13.0%) patients with CTCs non-detectable in the baseline sample had CTCs detectable in the Post-1 sample (Appendix A). Ten (37.0%) patients had exclusively CK^+^/VIM^+^/CD45^-^ (*n* = 5) or CK^+^/VIM^-^/CD45^-^ (*n* = 5) CTCs, whereas three additional patients harbored both CTC subpopulations (Table 2). There were no significant differences in the frequency of the various CTC subpopulations between the baseline and the Post-1 samples (*p* = 0.065; Wilcoxon test). 

ctDNA in the Post-1 was significantly decreased compared to baseline, as *EGFR* mutations were detected in the Post-1 sample in only 13 (27.7%) of the 47 patients who had LB status assessed in the study (*p* < 0.001). The *EGFR* T790M mutation was cleared in the Post-1 sample in all but one patient (Table 3). In 17 (36.2%) patients, there was a molecular ctDNA response with the disappearance of the *EGFR* mutation in the Post-1 sample, whereas for 1 (2.1%) patient the *EGFR* mutation was non-detected in the baseline sample but appeared in the Post-1 sample (Appendix A).

There was no correlation between the detection of CTCs and ctDNA in the post-Cycle 1 samples, since 6 (22.2%) patients had detectable ctDNA in their plasma, but not CTCs in the same blood drawn (*p* = 0.420).

### 3.5. CTC and ctDNA Status at the End of Study Treatment (EOT Sample)

In 34 (72.3%) patients, CTCs were measured at the EOT sample and 18 (52.9%) patients had detectable CTCs (total number of CTCs = 37 with a median number of 3/mL of blood; range, 1–6). Fifteen of these patients had available matched Post-1 and EOT samples: for one patient CTCs, were detectable Post-1, but were non-detectable at the EOT, whereas five (33.3%) patients with no detectable CTCs Post-1 had CTCs detectable at the EOT (Appendix A). Triple IF staining in 13 of these patients with available ISET filters revealed that CK^+^/VIM^+^/CD45^-^ and CK^+^/VIM^-^/CD45^-^ CTCs were exclusively detected in four and five patients, respectively, whereas in four additional patients, both subpopulations of CTCs were present (Table 2). 

The EOT results of ctDNA molecular profiling was available for 39 (83.0%) patients and *EGFR* mutations were detected in 24 (61.5%) patients; the detected *EGFR* mutations in the EOT samples are presented in Table 3; 12 (30.8%) patients with undetectable ctDNA in the Post-1 sample turned positive in the EOT sample (Appendix A). There was no correlation between the detection of CTCs and ctDNA in the end of treatment samples, since 7 (33.3%) patients had detectable ctDNA in their plasma but not CTCs in the same blood collection (*p* = 0.336).

### 3.6. Changes in LB Status during Treatment 

Considering that the detection of CTCs or/and ctDNA represent a positive LB status, the changes in LB status during treatment with osimertinib were further assessed. One treatment cycle turned negative the LB status in 18 (38.3%) patients, whereas 3 (6.4%) patients with negative LB status at baseline turned positive in Post-1 sample; similarly, 3 (7.7%) patients with positive LB status post-Cycle 1 turned negative at the EOT (Appendix A).

### 3.7. Clinical Outcome according to CTC, ctDNA and LB Status

The clinical response in terms of ORR and DCR was not associated with the detection of CTCs, ctDNA, or LB status at baseline or Post-1. Similarly, the detection of CTCs at any time point was not associated with PFS and OS (data not shown). Conversely, the median PFS was significantly shorter in patients with detectable ctDNA at baseline [6.0 vs. 15.9 months; 95% C.I., 3.8–8.3 vs. 8.6–23.3; *p* = 0.012) (Figure 1) and Post-1 (2.8 vs. 9.3 months; 95% C.I., 1.4–4.1 vs. 4.4–14.2, *p* < 0.001) (Table 4). The positive LB status at baseline was associated with shorter PFS (6.0 vs. 18.6 months; 95% C.I., 3.8–8.2 vs. 15.0–22.2, *p* = 0.020), but this was not the case for the positive LB status at Post-1 (*p* = 0.056) (Table 4 and Appendix A). The median OS was significantly shorter in patients with detectable ctDNA at baseline (13.5 vs. 30.9 months; 95% C.I., 10.4–16.6 vs. 14.6–47.2, *p* = 0.005), Post-1 (4.7 vs. 22.0 months; 95% C.I., 1.5–8.0 vs. 13.7–30.2, *p* < 0.001), and the EOT (6.0 vs. 12.0 months; 95% C.I., 0–12.6 vs. 9.2–14.8, *p* = 0.003) (Table 4; Figure 2A–C). The positive LB status was not associated with OS at any time point (Table 4). 

### 3.8. CTCs and ctDNA Changes during Treatment and Clinical Outcome

The monitoring of ctDNA during treatment revealed that patients with detectable ctDNA at baseline and Post-1 [Pre (+)/Post-1 (+)] experienced a significantly shorter PFS compared to patients without ctDNA at both time points [Pre (−)/Post-1 (−)] (2.3 vs. 15.9 months, *p* < 0.001; Table 5 and Figure 3A); in addition, patients with detectable ctDNA at baseline and Post-1 [Pre (+)/Post-1 (+)] had significantly shorter PFS compared to patients who turned ctDNA negative post Cycle 1 [Pre (+)/Post-1 (−)] (2.3 vs. 8.0 months, *p* < 0.001; Table 5 and Figure 3B). Furthermore, the PFS of patients with detectable ctDNA Post-1 and at the EOT [Post-1 (+)/EOT (+)] was significantly shorter compared to patients without detectable ctDNA at both time points [Post-1 (−)/EOT (−)] (2.3 vs. 8.9 months, *p* < 0.001; Table 5 and Figure 3C) as well as compared to those with non-detectable ctDNA Post-1 but with detectable ctDNA at the EOT [Post-1 (−)/EOT (+)]: (2.3 vs. 7.7 months, *p* < 0.001; Table 5 and Figure 3D). For OS, similar associations between ctDNA status at baseline and Post-1 were observed [Pre (+)/Post-1 (+) vs. Pre (−)/Post-1 (−): 4.2 vs. not reached; *p* < 0.001] and [Pre (+)/Post-1 (+) vs. Pre (+)/Post-1 (−): (4.2 vs. 19.8 months, *p* < 0.001; Table 5 and Figure 4A,B). Moreover, the median OS of patients with a ctDNA Post-1 (+)/EOT (+) was significantly shorter compared to patients with ctDNA Post-1 (−)/EOT (−): (4.2 vs. 26.3 months, *p* < 0.001; Table 5 and Figure 4C). In addition, patients with ctDNA Post-1 (+)/EOT (+) had significantly shorter OS compared to patients with non-detectable ctDNA Post-1 but with detectable ctDNA at the EOT [Post-1 (−)/EOT (+)]: (4.2 vs. 15.8 months, *p* < 0.001; Table 5 and Figure 4D).

### 3.9. Univariate and Multivariate Analyses 

The univariate analysis revealed that male sex and detection of ctDNA at baseline were significantly correlated with shorter median PFS (*p* = 0.002 and *p* = 0.015, respectively) and OS (*p* = 0.007 for both) (Table 6).

The multivariate analysis revealed gender (HR: 2.6; 95% C.I.: 1.2–5.5, *p* = 0.015) and ctDNA at baseline (HR: 3.0, 95% C.I.: 1.3–6.9, *p* = 0.009) as independent factors associated with PFS. For overall survival, only the detection of ctDNA at baseline emerged as independent factor associated with shorter OS (HR: 2.3; 95% C.I.: 1.1–5.1, *p* = 0.041), while only gender had a marked trend (HR: 2.0, 95% C.I.: 1.0–4.1, *p* = 0.05).

## 4. Discussion

We observed a significant decrease in ctDNA and CTCs after osimertinib treatment in EGFR-mutant NSCLC patients who had progressed on first-generation EGFR TKIs. Changes in ctDNA status between baseline and post-Cycle 1 emerged as an important and independent factor associated with clinical outcome. Although the heterogeneity of CTCs has been previously described in breast cancer patients [29,30], our study is the first to describe the heterogeneous phenotype of CTCs in EGFR-mutant NSCLC. 

In a previous study focused on a subgroup of the patients enrolled in the current study, we investigated the mRNA expression of epithelial, mesenchymal/EMT, and stem cell markers of CTCs and found a high prevalence of VIM-positive CTCs, indicating their EMT status during osimertinib treatment [37]. In addition, in the current study, we demonstrated that osimertinib resulted in a significant decrease of the CTCs after one month of treatment compared to pre-treatment values (*p* = 0.037). This is consistent with previous studies indicating that the number of CTCs can be used for monitoring drug efficacy [38,39]. Although the number of patients harboring CK^+^/VIM^+^/CD45^-^ decreased post-Cycle 1 and increased at the end of treatment, the observed differences were not statistically significant, probably due to the low number of patients in each group. Nevertheless, this seems to support our previous observation that osimertinib has no effect on the expression of mesenchymal markers [37]. 

The molecular ctDNA response has been associated with longer PFS to first-line EGFR TKIs [40]. In the present study, the ctDNA status during treatment with osimertinib was significantly correlated with the clinical outcome. ctDNA detection at baseline emerged as an independent factor associated with significantly shorter PFS and OS. The disappearance of ctDNA post-Cycle 1 was associated with longer PFS and OS, while the number of patients with detectable ctDNA increased at the end of treatment. Furthermore, among the 10 patients with detectable T790M in ctDNA at baseline, only 1 continued harboring T790M in ctDNA post-Cycle 1 and at the end of treatment, underlying the known effect of osimertinib on T790M-positive tumor cells [34]. Intriguingly, at the time of data analysis, four patients had not progressed on osimertinib for more than 3 years and all four had undetectable ctDNA at baseline and post-Cycle 1. These findings reinforce the clinical utility of the molecular ctDNA response [41]. In addition, the monitoring of ctDNA during osimertinib treatment could reveal patients with refractory or resistant oligometastatic disease who may benefit from radiation therapy, as has been shown by Wang et al. [42]. 

Our study did not confirm the previously reported predictive role of more than 5 CTCs per 7.5 mL of blood for osimertinib treatment [43]. We were also not able to find any correlation between the detection of CTCs and ctDNA, which implies that they represent distinct biologic phenomena. Whether CTC count and ctDNA could have a complementary prognostic value requires a larger cohort of patients, or even closer monitoring with liquid biopsies than in our study.

## 5. Conclusions

In conclusion, the CTCs and ctDNA detected were affected by osimertinib treatment in EGFR-mutant NSCLC patients, indicating a role of these blood biosources in addressing and monitoring drug efficacy. 

## Figures and Tables

**Figure 1 cancers-14-01574-f001:**
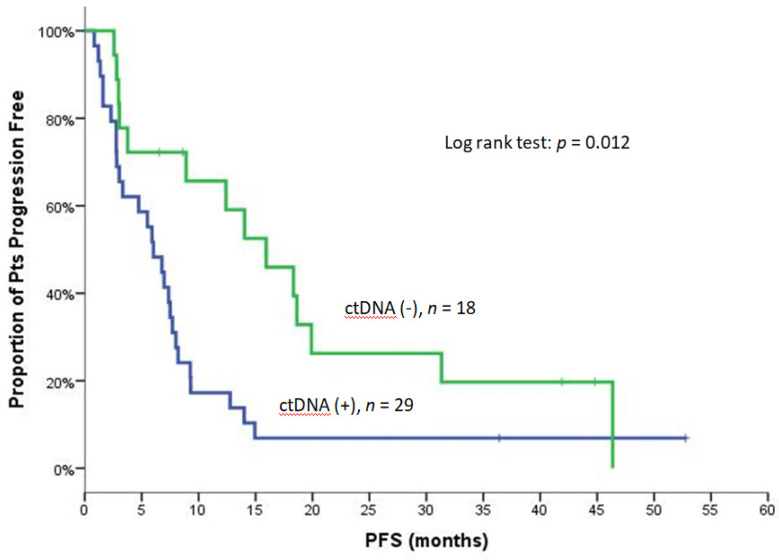
PFS for patients with detectable and non-detectable ctDNA at baseline.

**Figure 2 cancers-14-01574-f002:**
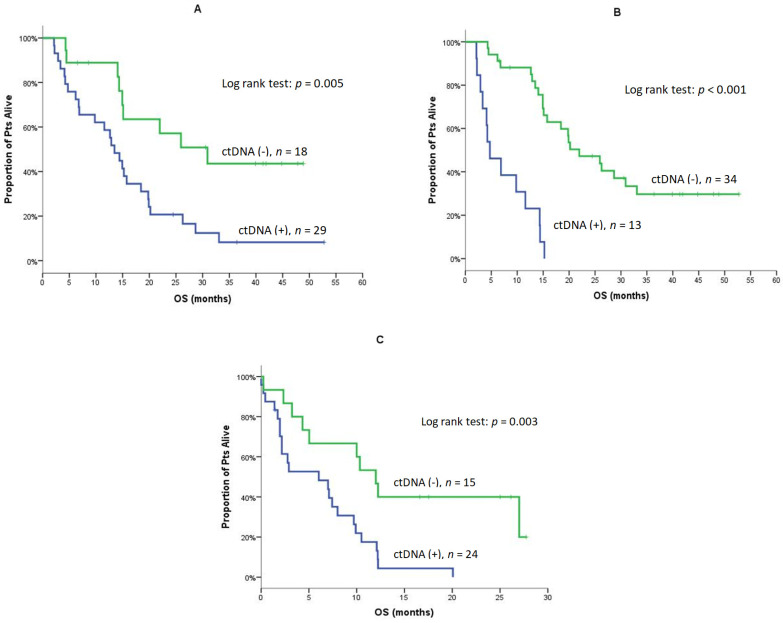
OS for Patients with detectable and non-detectable ctDNA at baseline (**A**), Post-1 (**B**) and, at the EOT (**C**).

**Figure 3 cancers-14-01574-f003:**
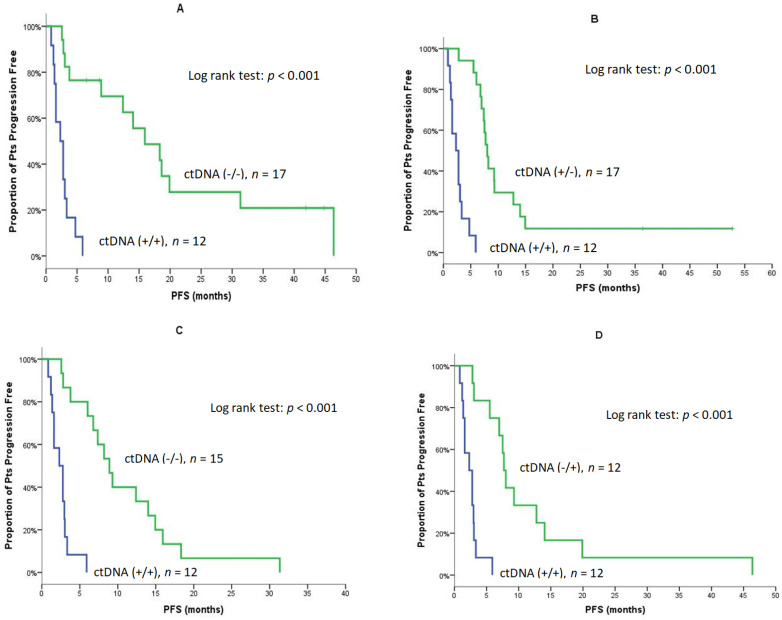
PFS of patients according to changes of ctDNA at Baseline, Post-1 and, at the EOT (**A**). Median PFS of patients with detectable ctDNA at Baseline and Post-1 [Pre (+)/Post-1 (+)] compered to those without detectable ctDNA at both time points [Pre (−)/Post-1 (−)]. (**B**) Median PFS of patients with detectable ctDNA at Baseline and Post-1 [Pre (+)/Post-1 (+)] compared to those with detectable ctDNA at baseline and non-detactable ctDNA Post-1 [Pre (+)/Post-1 (−)]. (**C**) Median PFS of patients with detectable ctDNA Post-1 and at the EOT [Post-1 (+)/EOT (+)] compared to patients without detectable ctDNA at both time points [Post-1 (−)/EOT (−)]. (**D**) Median PFS of patients with detectable ctDNA Post-1 and at the EOT [Post-1 (+)/EOT (+)] compared to patients without detectable ctDNA at Post-1 and detectable ctDNA at the EOT [Post-1 (−)/EOT (+)].

**Figure 4 cancers-14-01574-f004:**
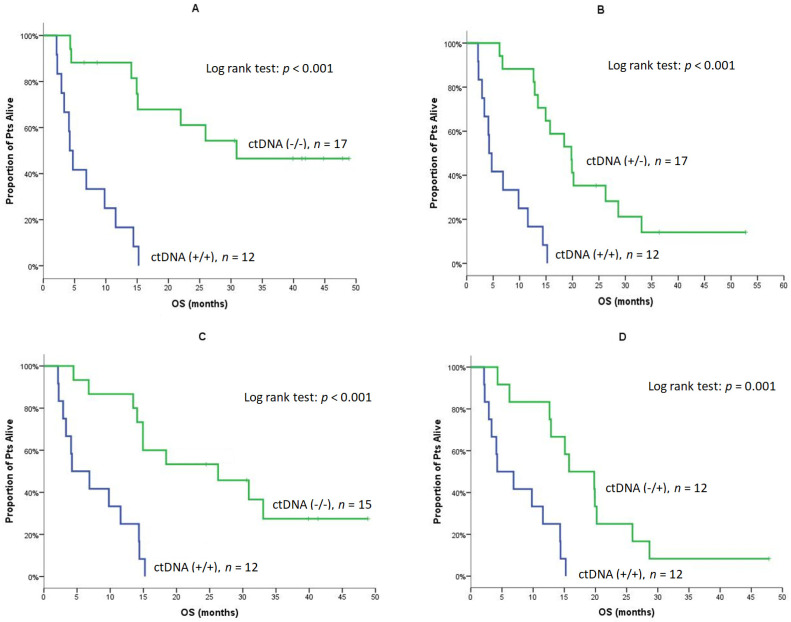
OS of patients according to changes of ctDNA at Baseline, Post-1 and at the EOT (**A**). Median OS of patients with detectable ctDNA at Baseline and Post-1 [Pre (+)/Post-1 (+)] compered to those without detectable ctDNA at both time points [Pre (−)/Post-1 (−)]. (**B**) Median OS of patients with detectable ctDNA at Baseline and Post-1 [Pre (+)/Post-1 (+)] compared to those with detectable ctDNA at baseline and non-detactable ctDNA Post-1 [Pre (+)/Post-1 (−)]. (**C**) Median OS of patients with detectable ctDNA Post-1 and at the EOT [Post-1 (+)/EOT (+)] compared to patients without detectable ctDNA at both time points [Post-1 (−)/EOT (−)]. (**D**) Median OS of patients with detectable ctDNA Post-1 and at the EOT [Post-1 (+)/EOT (+)] compared to patients without detectable ctDNA at Post-1 and detectable ctDNA at the EOT [Post-1 (−)/EOT (+)].

**Table 1 cancers-14-01574-t001:** Clinical characteristics and clinical outcome.

*N* = 47	*N* (%)
Age->Median (Min–Max)	66.0 (43–87)
Sex	
Male	13 (27.7)
Female	34 (72.3)
PS	
0	27 (57.4)
1	20 (42.6)
T790M (tissue and/or plasma) mutations	
Detected	17 (36.2)
Not detected	30 (63.8)
Line osimertinib admin	
Second Line	24 (51.1)
>Second Line	23 (48.9)
Previous Treatment	
Chemo and TKI	23 (48.9)
TKI only	24 (51.1)
Site of Disease	
Lung	39 (83.0)
LNs	24 (51.1)
Pleura	11 (23.4)
Liver	13 (27.7)
Bones	21 (44.7)
CNS	12 (25.5)
Other	9 (19.1)
Median sites involved	3 (1–5)
Response to osimertinib	
Complete response (CR)	2 (4.3)
Partial response (PR)	14 (29.8)
Stable disease (SD)	20 (42.6)
Progressive disease (PD)	11 (23.4)
ORR, 95% C.I	16 (34.0%; 20.5–47.6%)
DCR, 95% C.I	36 (76.6%; 64.5–88.7%
Relapses	41 (87.2)
PFS	
Median (mo; min–max)	7.5 (0.8–52.8)
95% C.I	6.0–9.0
Deaths	35 (74.5)
OS	
Median (mo; min–max)	15.1 (2.1–52.8)
95% C.I	10.8–19.4
1-year OS	69.8%
Follow up	
Median (mo; min–max)	41.9 (2.1–52.8)

ORR: overall response rate; DCR: disease control rate; OS: overall survival.

**Table 2 cancers-14-01574-t002:** CTCs and phenotype status at baseline, after Cycle 1, and at the EOT.

Time Points	CTCs		Phenotype	
	CTCs Detected	CK+VIM+CD45-	CK+VIM-CD45-	CK+VIM-CD45-/CK+VIM+/CD45-
Baseline (*n* = 32)	22 (68.8%)	8	4	10
Post 1st cycle (*n* = 27)	13 (48.1%)	5	5	3
EOT (*n* = 21)	13 (61.9%)	4	5	4

**Table 3 cancers-14-01574-t003:** Detection of ctDNA at each time point.

Detectable ctDNA	Baseline (*n* = 47)	Post-1 (*n* = 47)	EOT (*n* = 39)
ctDNA (at least one)	29 (61.7)	13 (27.7)	24 (61.5)
* T790M	10 (21.3)	1 (2.1)	1 (2.6)
Del19	18 (38.3)	8 (17.0)	11 (28.2)
L858R	8 (17.0)	4 (8.5)	9 (23.1)
S768I & G719X	3 (6.4)	1 (2.1)	3 (7.7)
H773_V774insNPH	-	-	1 (2.6)
C797S	-	-	1 (2.6)

* T790M mutation co-existed with other exon 19 and 21 EGFR mutations.

**Table 4 cancers-14-01574-t004:** PFS and OS based on ctDNA and LB status.

**PFS**	**Baseline (*n* = 47)**					
	**ctDNA (+)**	**ctDNA (−)**	** *p* **	**Liquid (+)**	**Liquid (−)**	** *p* **
N/Events	29/27	18/14		38/35	9/6	
Median	6.0	15.9	**0.012** (Figure 1)	6.0	18.6	**0.020** (Appendix A)
Min–Max	0.8–52.8	2.6–46.4		0.8–52.8	3.0–44.8	
95% C.I.	3.8–8.3	8.6–23.3		3.8–8.2	15.0–22.2	
	**Post-1 (*n* = 47)**					
	**ctDNA (+)**	**ctDNA (−)**	** *p* **	**Liquid (+)**	**Liquid (−)**	** *p* **
N/Events	13/13	34/28		23/22	24/19	
Median	2.8	9.3	**<0.001**	3.3	8.2	0.056
Min–Max	0.8–5.9	2.6–52.8		0.8–52.8	2.6–44.8	
95% C.I.	1.4–4.1	4.4–14.2		0.6–6.1	5.8–10.6	
**OS**	**Baseline (*n* = 47)**					
	**ctDNA (+)**	**ctDNA (−)**	** *p* **	**Liquid (+)**	**Liquid (−)**	** *p* **
N/Events	29/26	18/9		38/30	9/5	
Median	13.5	30.9	**0.005** (Figure 2A)	14.9	25.9	0.093
Min–Max	2.1–52.8	4.3–48.9		2.1–52.8	6.5–44.8	
95% C.I.	10.4–16.6	14.6–47.2		12.3–17.5	13.5–38.3	
1-year OS	58.6%	88.9%		62.9%	100%	
	**Post-1 (*n* = 47)**					
	**ctDNA (+)**	**ctDNA (−)**	** *p* **	**Liquid (+)**	**Liquid (−)**	** *p* **
N/Events	13/13	34/22		23/18	24/17	
Median	4.7	22.0	**<0.001** (Figure 2B)	14.3	20.2	0.269
Min–Max	2.1–15.2	4.3–52.8		2.1–52.8	4.3–48.9	
95% C.I.	1.5–8.0	13.7–30.2		9.9–18.8	14.9–25.5	
1-year OS	23.1%	88.1%		56.5%	83.1%	
	**EOT (*n* = 39)**					
	**ctDNA (+)**	**ctDNA (−)**	** *p* **	**Liquid (+)**	**Liquid (−)**	** *p* **
N/Events	24/23	15/10		29/25	10/8	
Median	6.0	12.0	**0.003** (Figure 2C)	7.1	10.0	0.250
Min–Max	0–20.1	0.2–27.7		0–27.7	0.2–27.0	
95% C.I.	0–12.6	9.2–14.8		1.3–12.9	0–20.8	
1-year OS	17.5%	46.7%		25.1%	40.0%	

*p*-values marked with bold indicate statistically significant.

**Table 5 cancers-14-01574-t005:** PFS and OS according to changes in ctDNA at Baseline, Post-1, and the EOT.

**PFS**	**Baseline/Post-1**					
	**+/+**	**−/−**	** *p* **	**+/+**	**+/−**	** *p* **
N/Events	12/12	17/13		12/12	17/15	
Median	2.3	15.9	**<0.001** (Figure 3A)	2.3	8.0	**<0.001** (Figure 3B)
Min–Max	0.8–5.9	2.6–46.4		0.8–5.9	2.8–52.8	
95% C.I.	1.0–3.6	8.1–23.7		1.0–3.6	7.0–8.9	
**PFS**	**Post-1/EOT**					
	**+/+**	**−/−**	** *p* **	**+/+**	**−/+**	** *p* **
N/Events	12/12	15/15		12/12	12/12	
Median	2.3	8.9	**<0.001** (Figure 3C)	2.3	7.7	**<0.001** (Figure 3D)
Min–Max	0.8–5.9	2.6–31.3		0.8–5.9	2.8–46.4	
95% C.I.	1.0–3.6	6.5–11.3		1.0–3.6	6.9–8.5	
**OS**	**Baseline/Post-1**					
	**+/+**	**−/−**	** *p* **	**+/+**	**+/−**	** *p* **
N/Events	12/12	17/8		12/12	17/14	
Median	4.2	NE	**<0.001** (Figure 4A)	4.2	19.8	**<0.001** (Figure 4B)
Min–Max	2.1–15.2	4.3–48.9		2.1–15.2	6.2–52.8	
95% C.I.	3.1–5.3	-		3.1–5.3	14.2–25.3	
1-year OS	16.7%	88.2%		16.7%	88.2%	
**OS**	**Post-1/EOT**					
	**+/+**	**−/−**	** *p* **	**+/+**	**−/+**	** *p* **
N/Events	12/12	15/10		12/12	12/11	
Median	4.2	26.3	**<0.001** (Figure 4C)	4.2	15.8	**0.001** (Figure 4D)
Min-Max	2.1–15.2	4.5–48.9		2.1–15.2	4.3–47.8	
95% C.I.	0–8.9	8.8–43.8		0–8.9	7.8–23.7	
1-year OS	25.0%	86.7%		25.0%	83.3%	

*p*-values marked with bold indicate statistically significant.

**Table 6 cancers-14-01574-t006:** Univariate for PFS and OS.

Independent Factors	PFS		OS	
	Hazard Ratio (95% Confidence Interval)	*p*	Hazard Ratio (95% Confidence Interval)	*p*
**Performance status**				
PS 0	1 (reference)		1 (reference)	
PS 1	1.2 (0.7–2.3)	0.534	1.8 (0.9–3.5)	0.094
**Gender**				
Male	3.2 (1.5–6.6)	**0.002**	2.6 (1.3–5.2)	**0.007**
Female	1 (reference)		1 (reference)	
**Line of osimertinib**				
Second line	1.0 (0.5–1.7)	0.835	1.1 (0.6–2.2)	0.751
>Second line	1 (reference)		1 (reference)	
**T790M (plasma and/or tissue)**				
Detected	1.0 (0.5–2.0)	0.886	1.0 (0.5–2.0)	0.955
Not detected	1 (reference)		1 (reference)	
**Baseline ctDNA**				
Detected	2.3 (1.2–4.5)	**0.015**	2.9 (1.3–6.2)	**0.007**
Not detected	1 (reference)		1 (reference)	

*p*-values marked with bold indicate statistically significant.

## Data Availability

The data generated in this study are available upon reasonable request from the corresponding author.

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
