# Peer review of "Effect of Osimertinib on CTCs and ctDNA in EGFR Mutant Non-Small Cell Lung Cancer Patients: The Prognostic Relevance of Liquid Biopsy"

_cancers, 2022, doi:10.3390/cancers14061574_

Round 1

Reviewer 1 Report

The present manuscript reports on the predictive role of early CTCs/ctDNA clearance during osimertinib and outcome in advanced NSCLC patients harboring EGFR mutations.

Comments:

  • Patients with inadequate ctDNA response might benefit from addition of chemotherapy and/or other therapies. Recently, addition of radiation therapy has been reported to improve the outcomes of TKIs in oligometastatic NSCLC (Wang XS, et al. J Natl Cancer Inst. 2022). Can these findings better assist patients’ selection to this (and other combinatorial) approach?
  • The presence of concomitant mutations (TP53, RB1, etc.) has been reported to be associated with inferior outcomes with EGFR TKIs (Aggarwal C, et al. JCO Precis Oncol 2018). Did the authors analyze these mutations in tissue samples?
  • According to Suppl. Table 1, three patients harbored a G719X + S768I mutation and one patient harbored an exon 20 insertion. Detection of these mutations on ctDNA is missed when using some PCR-based tests. Indeed, Table 3 reports that three patients had “other mutations” ctDNA detection at baseline. Please clarify and provide additional details on mutations detected with the test used here.
  • Figures should be uploaded in a higher quality format.

Author Response

  • We thank the reviewer for his relevant comment. We believe that the monitoring of the ctDNA during Osimertinib could be used in the real life as a tool for stratify patients according to their risk for drug resistance and, thus to early implementation of radiotherapy in patients with oligometastatic disease. We added this option in the discussion section (page 19, 1st paragraph) and the corresponding reference was added (#42)
  • We have not performed a polygenic NGS analysis for the molecular profiling of the primary tumors since this approach was not a standard of care in several of the participating centers and a central molecular analysis of the primary tumors was not an inclusion criterion. Therefore, we have no data regarding the presence of concomitant mutations.
  • Table 3 was completed with the detected mutation in the cases which have been mentioned as “other mutations” in the initial manuscript
  • A new set of Figures of better quality have been submitted with the revised manuscript. 

Reviewer 2 Report

In the present manuscript the authors have established the implications of liquid biopsy in the assessment of treatment efficacy. They have conducted a comprehensive study on EGFR mutant NSCLC patients and treated with Osimerteinib.

The authors have shown, positive change among the CTCs due to Osimertinib treatment.

They have nicely wrote the manuscript and explained the result. It will be good if they can improve the quality of the figures.

Author Response

We thank for the reviewer’s positive comments. A new set of Figures of better quality have been submitted with the revised manuscript. 

Reviewer 3 Report

1) It seems to me that the name is not quite correct. More appropriate would be "Prognostic value of CTCs and ctDNA in the treatment of Osimertinib in EGFR mutant Non-Small Cell Lung Cancer Patients". This is as an example. 2) Why is the change in ctDNA (+/+) and (-/-) not taken into account in multivariate analysis. According to Figures 3 and 4, the difference in survival rates is statistically significant. 3) A list of abbreviations should be given to table 1, it is not convenient to read, you have to look for a transcript in the text of the manuscript. 

Author Response

  • We thank the reviewer for his comment and we agree with him. According to his advise we added in the title the mention “the prognostic relevance of liquid biopsy”
  • According to the statistician of our group we cannot include in the same model both the ctDNA (pos vs neg) and that of pre/post status since they are “linearly dependent covariates” and therefore the model cannot accept this. In addition, the multivariate analysis using the pre/post status will include a lower number of patients leading to a larger range of 95% CI. Furthermore the small number of the different subgroups limits the possibility of further analysis with the multivariate model. Therefore, we preferred to perform the multivariate model using the ctDNA in the whole group of enrolled patients.
  • We have explained the abbreviations in the Table 1

Reviewer 4 Report

The article entitled “Effect of Osimertinib on CTCs and ctDNA in EGFR mutant Non-Small Cell Lung Cancer Patients” is an original research study using plasma from EGFR positive NSCLC patients treated with Osimertinib. This work is trying to show reliability of liquid biopsy as a predictive tool to judge treatment efficacy. The article is well written and concluded. However, some minor changes are required.

  • In figure 2, subfigure should be 2C instead of 3C and should be updated in figure legend.
  • The DOI is missing in reference #42.

Author Response

  • Figure 2 has been replaced with another one more relevant. Moreover, the legend has been changed
  • DOI ref 42 reformatted and now is ref 43.
  • Unfortunately ref 43 Yang, B et al hasn’t any DOI number. There is only PMID: 30358214

https://pubmed.ncbi.nlm.nih.gov/30358214/

Reviewer 5 Report

The manuscript is well written, figures are of poor quality and should be improved

Author Response

The figures were improved according to the reviewer’s comment.

Reviewer 6 Report

The manuscript by Kallergi et al., titled “Effect of Osimertinib on CTCs and ctDNA in EGFR mutant Non-Small Cell Lung Cancer Patients” has been reviewed. Overall, I find the manuscript interesting and suitable for publication in Cancers. I have two points that should be addressed before the manuscript can be accepted for publication.

First, the simple abstract should be re-written. It is too detailed and includes many abbreviations, which have not been written out.

Second, the text on the survival curves cannot be read and this problem should be fixed.

Author Response

  • The simple abstract was re-written.
  • The text of the survival curves was corrected.

Round 2

Reviewer 3 Report

I have no more comments on the article. In its present form, the article can be recommended for publication.